# How Did the COVID-19 Pandemic Affect Population Mobility in Taiwan?

**DOI:** 10.3390/ijerph191710559

**Published:** 2022-08-24

**Authors:** Shih-Feng Liu, Hui-Chuan Chang, Jui-Fang Liu, Ho-Chang Kuo

**Affiliations:** 1Department of Respiratory Therapy, Kaohsiung Chang Gung Memorial Hospital, Kaohsiung 833, Taiwan; 2Division of Pulmonary and Critical Care Medicine, Department of Internal Medicine, Kaohsiung Chang Gung Memorial Hospital, Kaohsiung 833, Taiwan; 3College of Medicine, Chang Gung University, Taoyuan 333, Taiwan; 4Department of Respiratory Care, Chang Gung University of Science and Technology, Chiayi 600, Taiwan; 5Chronic Diseases and Health Promotion Research Center, Chang Gung University of Science and Technology, Chiayi 600, Taiwan; 6Department of Paediatrics and Kawasaki Disease Center, Kaohsiung Chang Gung Memorial Hospital, Kaohsiung 833, Taiwan

**Keywords:** population mobility, COVID-19, pandemic, Taiwan, Alpha, Omicron

## Abstract

Background: Coronavirus disease (COVID-19) impairs the free movement of human beings. The study aims to determine how the COVID-19 pandemic affected population mobility. Methods: The study obtained Google COVID-19 population mobility report and e Taiwan COVID-19 pandemic information from Our World in Data. Results: During the Alpha wave, transit decreased the most, with an average difference of >50%, followed by parks, workplaces, groceries, and pharmacies. During the Omicron wave, the average population flow in parks and transit decreased by about 20%. During the pre-existing wave, the average population visits of transit decreased by 10% at the most, followed by parks and workplaces. The peak of daily new confirmed cases per million (7-day rolling average) was 25.02, 6.39, and 0.81 for Alpha, Omicron, and the pre-existing wave, respectively. Daily new confirmed cases per million people correlated with the change in population visits of various places (all *p* < 0.001). The reproduction rate (7-day rolling average) correlated with the change of population visits of most places, except retail and recreation. We conclude the Alpha variant affected more individuals than Omicron and pre-existing type. Furthermore, changes in population visits in transit were most impacted. This change was consistent with daily new confirmed cases per million people and reproduction rate (7-day rolling average). Conclusion: The Alpha variant affected more individuals than the Omicron and pre-existing types. Furthermore, changes in population visits in transit locations were most impacted. This change was consistent with the daily new number of confirmed cases per million people and the 7-day rolling average reproduction rate.

## 1. Introduction

The Coronavirus disease 2019 (COVID-19) pandemic was caused by the severe respiratory syndrome coronavirus 2 (SARS-CoV-2) [1]. The virus was first discovered in Wuhan, Hubei Province, China, in late 2019 [2], and then quickly spread globally in early 2020 [3]. As of 13 March 2022, the cumulative number of confirmed cases worldwide exceeded 470 million, and the number of deaths exceeded 6 million [4]. To date, the pandemic continues to spread, and the cumulative number of infected cases has also increased. SARS-CoV-2 is dominantly airborne [5]. The infection peaks the day before the symptoms appear and decreases as symptoms appear [6]. Unlike SARS-CoV, the SARS-CoV-2 infection may occur before fever and is more contagious than the SARS-CoV [7]. Furthermore, due to structural issues, SARS-CoV-2 has different variants, each with different infectivity, disease severity, and interactions with host immunity [8]. For example, the recent Omicron variants are highly contagious [9], and breakthrough infections are still possible in patients vaccinated with various vaccine combinations [10]. However, the proportion of critically ill patients has significantly decreased [11].

Due to the high infectivity of SARS-CoV-2, pandemic prevention measures in various countries include border quarantine, monitoring the TOCC (travel, occupation, contact, and cluster) medical history, wearing masks, washing hands frequently, taking body temperature, maintaining a safe social distance, and other related health policies [12,13]. The pandemic prevention policies in various countries have also been adjusted in a rolling manner according to pandemic severity. Looking back, since the outbreak, the exchanges between countries have decreased, and interaction between people has also declined. The pandemic has restricted the scope of people’s activities and changed the population mobility in various places.

Since the first case was discovered in Taiwan in January 2020, with the cooperation between the government and the people, its pandemic prevention performance has been praised by many countries. Although sporadic variants appear occasionally, the pandemic situation in Taiwan can be roughly divided into three main waves. The first wave is mainly the pre-existing variant, the second is the Alpha variant, and the third wave is the Omicron variant [14]. There is a strong correlation between the COVID-19 pandemic and population mobility [15,16,17,18]. Some may be in isolation due to the fear of patients in self-defense, or due to the government’s public health policy to prevent the spread of large-scale infections. Taiwan is an island country surrounded by sea and has natural border protection. It takes only 90 min to take the high-speed train from Taipei to Kaohsiung. The electronic media information system is very developed. During the epidemic, people have abided by the government’s public health policy and reduced visits to various places. There has been no phenomenon of migration. The purpose of this study is to explore the situation of population mobility in Taiwan during COCID-19 epidemic and to explore the correlation between population mobility and public health strategies, confirmed cases, and reproduction rate.

## 2. Methods

### 2.1. Study Design

We obtained data from the Google online COVID-19 mobility report [19] and Taiwan’s pandemic situation from Our World in Data [14] and from the Taiwan Center for Disease Control [12]. We then compared the trend of population visits in various venues in Taiwan during the three main pandemic waves: pre-existing, Alpha, and Omicron, and investigate the correlation between population mobility and public health strategies, confirmed cases, and reproduction rate. Changes in population visits in in various venues

The study aimed to elucidate the trend of people visits in various venues in the community due to COVID-19 considering the Google online COVID-19 community mobility report (www.google.com/COVID19/mobility/ (accessed on 1 April 2022)), including retail stores and leisure facilities (i.e., restaurants, cafes, shopping malls, theme parks, museums, libraries, movie theaters, and other similar places); grocery stores and pharmacies (i.e., grocery stores, farmers’ markets, specialty food stores, drugstores, pharmacies, and other similar places); mass transit stations (i.e., subway, bus, and train stations) such as mass transit transfer stations and other similar places; workplaces; and residential zones. Google community mobility report used data from a 5-week period (3 January to 6 February 2020) to calculate the median for each day of the week, and this was used as a benchmark to display the number of visitors and visitor stays at different locations based on this specific benchmark value. These changes were calculated and aggregated using de-identified data, subject to a strict privacy policy.

### 2.2. COVID-19 Pandemic in Taiwan

We obtained information on the COVID-19 pandemic from the Taiwan Center for Disease Control, Ministry of Health and Welfare (www.cdc.gov.tw (accessed on 1 April 2022)), and Our World in Data (ourworldindata.org/coronavirus (accessed on 1 April 2022)).

From the trend chart of daily new confirmed COVID-19 cases per million people (7-day rolling average), it was roughly divided into three main waves of the pandemic: pre-existing, Alpha, and Omicron. 

### 2.3. Study Population

The integrated data of confirmed COVID-19 cases was uploaded to Our World in Data by the Taiwan Center for Disease Control, from which an overview of statistical information such as daily new confirmed cases, confirmed deaths, reproduction rate, case fatality, and people vaccination and so forth, can be obtained. Personal information was de-identified.

### 2.4. Ethical Issues

We used information that was lawfully publicly available for publicly known purposes. This study was approved by the Human Experiment Ethics Committee of Chang Gung Memorial hospital and is in line with the exemption criteria (202000966B1).

### 2.5. Statistics

The changes in the people flow in various places from Google community mobility reports were calculated and expressed as absolute values and as percentages of the percent from baseline, including mean, median, and maximum during the three main pandemic waves. The new daily confirmed cases during the three waves are shown as the mean, median and maximum. The relationship between changes in the population of various places and 7-day rolling average reproduction rate/and daily new confirmed cases per million people (7-day rolling average) were analyzed using multiple regression analysis. Data analysis was conducted using STATA Version 12 (College Station, TX, USA). A two-sided *p*-value < 0.05 was considered statistically significant.

## 3. Results

### 3.1. Changes in the Population Flow at Each Place in the Pre-Existing, Alpha, and Omicron Waves 

Table 1 illustrates the mean, medium, and maximum number of daily new confirmed cases for the three pandemic waves in Taiwan, and the mean, medium, and maximum difference between the reported population flow value and the baseline value in each location. During the Alpha variant wave, the number of differences in the mean, medium, and maximum number of the population flow in each place was the largest, followed by the Omicron variant, and finally the pre-existing type. During the Alpha variant wave, transit recorded the greatest decrease, with an average difference of >50%, followed by parks, workplaces, groceries, and pharmacies, while the residential flow increased. During the Omicron variant wave, the average flow of people in parks and transit decreased by about 20%, while pharmacies and residential zones saw an increase. During the pre-existing wave, the average population flow of transit decreased by 10% at the most, followed by parks and workplaces, while the average population flow in groceries, pharmacies, and residential zones increased slightly, all within <1% (Figure 1).

### 3.2. The Relationship between Changes in the Population Visits for Various Locations and the Number of Daily New Confirmed Cases Per Million People (7-Day Rolling Average)

The peak of daily new confirmed cases per million people (7-day rolling average) during the Alpha variant pandemic reached 25.01, which was 6.39 for the Omicron variant and 0.81 for the pre-existing type. The number of COVID-19 daily new confirmed cases per million people (7-day rolling average) was significantly correlated with population visits changes in transit (*p* < 0.001), groceries and pharmacies (*p* < 0.001), residential zones (*p* < 0.001), retail and recreation (*p* < 0.001), parks (*p* < 0.001), and workplaces (*p* < 0.001) (Table 2 and Figure 2). 

### 3.3. The Relationship between Changes in the Population Visits of Various Locations and the Reproduction Rate (7-Day Rolling Average)

The peak of the reproduction rate (7-day rolling average) during the Alpha variant pandemic reached 2.06, which was 1.69 for the Omicron pandemic and 1.29 for the pre-existing pandemic.

The reproduction rate (7-day rolling average) was significantly correlated with changes in the population flow of transit (*p* < 0.001), groceries and pharmacies (*p* < 0.001), workplaces (*p* < 0.001), residential zones (*p* < 0.001), and parks (*p* = 0.04), but it was not significantly correlated with changes in population flow for retail and recreation (*p* = 0.3) (Table 3 and Figure 3).

## 4. Discussion

This study found that the COVID-19 pandemic has indeed affected the movement trends of Taiwan’s population in various locations. Among the three pandemic waves, the Alpha wave had the greatest impact on the changes in the population visits, followed by the Omicron variant, and finally the pre-existing type. Population flows in transit and parks declined more than elsewhere, while population visits in residential areas increased in all three waves, especially during the Alpha wave. Additionally, the number of daily new confirmed cases per million people (7-day rolling average) was correlated with the change in the population flow of various places. The reproduction rate (7-day rolling average) correlated with the change in the population visits of most places, except in retail and recreation.

The peak of daily new confirmed cases per million people (7-day rolling average), reproduction rate (7-day rolling average), and newly confirmed deaths per million people (7-day rolling average) during the Alpha wave was higher than that of the Omicron and the pre-existing wave. Therefore, the Taiwan Epidemic Command Center raised the epidemic prevention alert to the third level during the Alpha wave, resulting in the greatest reduction in the people visits in various places during this period, especially in transit. The epidemic prevention alert to the third level includes intensive infection control in mass transit; school closures or school closures; intensive infection control or closure of public places; rapid containment; sheltering; and domestic travel restricted area lockdown. Coupled with the law-abiding spirit of Taiwanese people, the frequencies of population visits of various places are the same as our data presentation. During the Alpha epidemic period, people could not travel, could not take public transportation, stopped work and classes, would go to public places as little as possible, would try not to go shopping, stay at home more, and sometimes go to the pharmacy to buy masks and disinfection supplies, so presented transit visits decreased by the largest amount, followed by visits to the park and workplace; time spent at home increased, and there was not much difference recorded in pharmacy visits. During the Omicron epidemic, the government policy was lowered to a second-level alert; thus, the frequencies of visits to various places ranked the second. During the epidemic period of the pre-existing variant, the epidemic situation was the lightest, and the frequencies of visits to various places was the lowest. The Omicron variant of COVID-19 is currently circulating in Taiwan. It entered Taiwan in January 2022. In the literature, it is characterized by high infectivity [20], and breakthrough infections even after vaccination [21,22]. The Omicron variant may contain multiple mutations that mediate immune evasion [23]. However, up to January 2022, the percentage of the population in Taiwan that received at least one dose of vaccination had reached approximately 80% and that of the fully vaccinated was 65%, the Omicron variant cases are mostly mild, and serious illness is rare. The national epidemic prevention policy is gradually moving toward coexisting with the virus, but changes in the epidemic situation are still closely monitored and rolling adjustments have been made. At present, the epidemic prevention alert remains at Level 2, and epidemic prevention is mainly based on wearing masks and maintaining social distance. Therefore, changes in the people visits in various places during the Omicron wave lay between the Alpha wave and the pre-existing type.

COVID-19 has indeed caused many countries to take urgent public health measures, and limiting people’s population mobility is the most common and possible and effective mitigation strategy [15,16,17,18]. However, there may be some adverse social and economic consequences for the most vulnerable groups [17]. However, there are also studies that suggest that there are significant differences in the effectiveness of policies between different regions with different population mobility patterns [18]. For example, outflow mobility restrictions were ineffective in reducing death cases in population influx areas, and restrictions on inflow mobility (or intra-city mobility) were ineffective in reducing confirmed cases (or death cases) in population outflow areas [18]. These may require further integration and discussion by public health experts. Due to the special geographical environment of an island country, Taiwan has an excellent electronic communication system, and has a law-abiding people. It also provides us with valuable experience, which can be compared with countries in different situations.

COVID-19 affects the trend of population visits in various locations [20]. In addition to different SARS-CoV-2 variants, the vaccination coverage rate, government control alert level, people cooperation, reproduction rate, daily new confirmed cases, and mortality rate are also important factors. In the future, the inventions of more effective vaccines and new antiviral drugs will also lead to changes in the national policy and the current state of population mobility.

There are several limitations to this study. First, the population movement was collected from Google’s online community mobility reports, which do not represent the entire population movement in Taiwan. Second, the three waves of the pandemic are divided into the pre-existing, Alpha, and Omicron waves. This is mainly a general classification, which cannot completely cover a small number of variants. Third, there may be a slight difference in the synchronization between the epidemic statistics of Our World in Data and the actual situation in Taiwan, but it does not affect the general situation. Fourth, we only used Our World in Data up until 1 February 2022, and the number of new confirmed Omicron cases is still on the rise.

## 5. Conclusions

COVID-19 is an infectious disease prevalent globally, and it does affect changes in the population mobility in Taiwan. The changes in population visits of various places during the Alpha wave were greater than that in the Omicron wave and the pre-existing type. During the COVID-19 pandemic, the reduction of population visits in transit was greater than that in other places. The daily new confirmed cases per million people (7-day rolling average) and the reproduction rate (7-day rolling average) were consistent with the changes in the population visits in various places.

## Figures and Tables

**Figure 1 ijerph-19-10559-f001:**
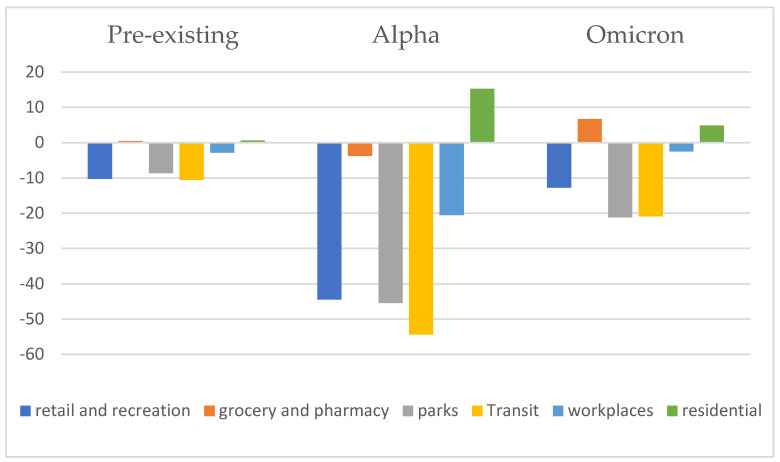
Taiwan population flow change of various places in the community due to COVID-19 epidemic.

**Figure 2 ijerph-19-10559-f002:**
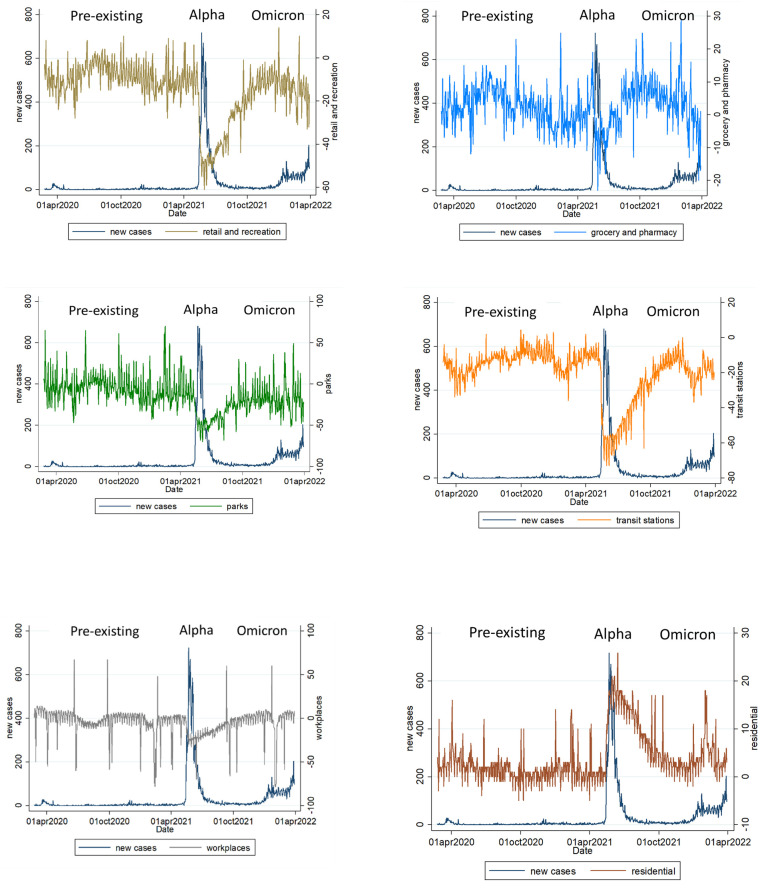
Population visits change of various places well correlated with daily new confirmed cases per million people (7-day rolling average) in Taiwan during the main three wave COVID-19 epidemics.

**Figure 3 ijerph-19-10559-f003:**
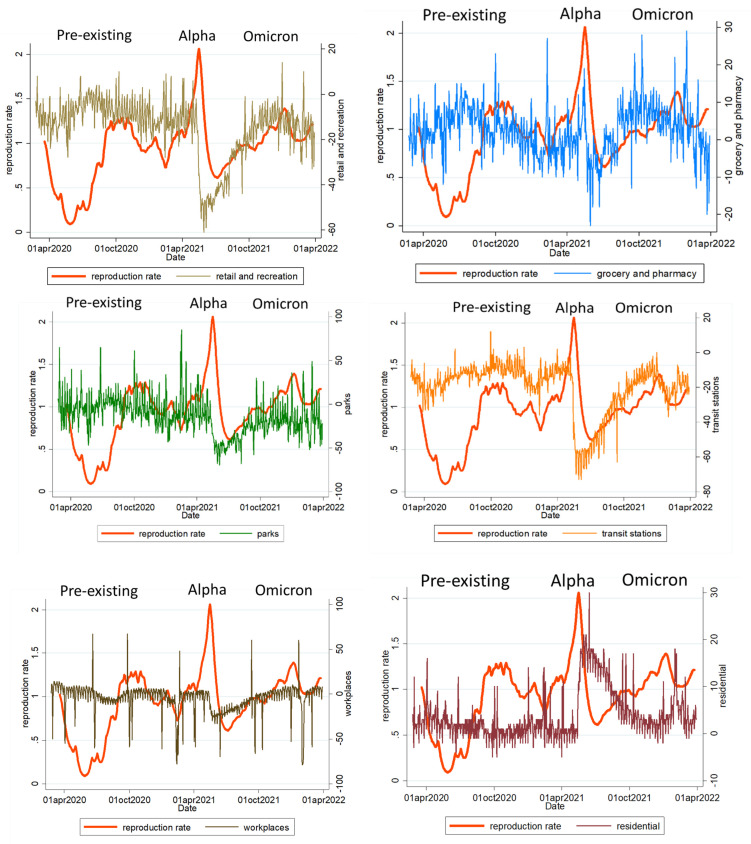
The relationship between changes in the population visits of various places and the 7-day rolling average reproduction rate in Taiwan during the main three waves COVID-19 epidemics.

**Table 1 ijerph-19-10559-t001:** The population flow change of each place in the Pre-existing epidemic, Alpha epidemic, and Omicron epidemic in Taiwan.

	Pre-Existing			Alpha			Omicron		
	**Average**	**Medium**	**Max**	**Average**	**Medium**	**Max**	**Average**	**Medium**	**Max**
New case	4.39	4	25	217.37	175	723	56.91	57.5	129
retail and recreation	−10.288	−12	7	−44.51	−46	−61	−12.77	−11	−27
grocery and pharmacy	0.48	1	−11	−3.82	−4	−23	6.67	4.5	−6
Parks	−8.71	−13	−38	−45.46	−45.73	−70	−21.22	−22	−50
Transit stations	−10.61	−11	−18	−54.4	−56	−73	−20.94	−20	−37
Workplaces	−2.90	1	−59	−20.58	−22	−70	−2.56	3	−79
Residential	0.60	1	−5	15.26	16.5	−1	4.84	4	0

**Table 2 ijerph-19-10559-t002:** Relationship between changes in the population flow of various locations and daily new confirmed cases per million people (7-day rolling average).

	Coef.	95% CI	*p*-Value
Retail and recreation	−1.233	−2.594~0.128	0.076
Grocery and pharmacy	−1.332	−2.327~−0.338	0.009
Park	0.277	−0.177~0.731	0.231
Transit stations	−1.397	−2.467~−0.328	0.011
workplaces	0.250	−0.282–0.781	0.357
residential	3.882	1.259–6.504	0.004

**Table 3 ijerph-19-10559-t003:** Relationship between changes in the population flow of each place and reproduction rate (7-day rolling average).

	Coef.	95% CI	*p*-Value
Retail and recreation	−0.023	−0.044~−0.002	0.031
Grocery and pharmacy	0.007	−0.008~0.023	0.357
Park	−0.005	−0.013~0.002	0.150
Transit stations	0.023	0.006~0.040	0.008
Workplaces	−0.008	−0.017~0.000	0.062
Residential	−0.055	−0.240~0.130	0.517

## Data Availability

The data generated and analyzed in this study are included in this article. The data were obtained from the Google online COVID-19 community mobility report (www.google.com/covid19/mobility/ (accessed on 1 April 2022)) and our world in data (ourworldindata.org/coronavirus (accessed on 1 April 2022)). The datasets used and/or analysed during the current study are available from the corresponding author on reasonable request.

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
