# Peer review of "How Did the COVID-19 Pandemic Affect Population Mobility in Taiwan?"

_ijerph, 2022, doi:10.3390/ijerph191710559_

Round 1

Reviewer 1 Report

1. Since the strong correlation between population mobility and the COVID-19 pandemic has been proved many times by existing researches, the introduction part needs to be strengthened to clarify the research question and its uniqueness.

2. “community mobility” is the keyword here, but the paper doesn’t explain whether it is intra-community mobility or community-based mobility. In addition, how does the data used in the paper reflect the community mobility?

3. Literature review about epidemics prevention policies has little to do with the research question of the paper. It is necessary to summarize the existing results of the COVID-19 pandemic on community mobility.

4. The “ method” part needs to be reorganized to clarify the research design, the source of data, and the ethical issues.

5. Spearman correlation is too weak to demonstrate how and to what extent the COVID-19 pandemic is affecting community mobility. A mathematical model can be further constructed for analysis.

6. The data-based findings of the paper are not strongly related to the proposed research question. There is a lack of discussion on the trend change (spatial mobility) of population mobility in different places and the reasons for that change.

Reviewer 2 Report

The articles analyses impacts of different COVID-19 waves especially on on transit usage and location type.

It's pretty interesting to see how Taiwan reacted to the 3 CODIV waves. I think it could be interesting to see these results compared with other cities, analyzing either other publications or data.

I don't like the term 'population flow' - I think you're referring to activity flows/visits.

I'd be better to emphasize a little more about differences between location type in terms of how they reacted to the COVID: what location type reacted more, with what magnitude, and what sign (positive correlated or negative correlated), which I think it's the key point of the paper. In particular I'd try do to give some explanations to these numbers.

I'd also better discuss the implication of this study: how you think better comprehending these trends could help future research but not only.

Reviewer 3 Report

The manuscript is well prepared except for few issues that need to be addressed.

1. The significance/uniqueness of the manuscript is missing in the intro section. The last paragraph should do justice to that.

2. There is the need to explain why the authors used data from "5-week period (January 3 to February 6, 2020)." Why the narrow range? Is there anything special about this window?

3. Please explain the quality of your data, sample characteristics, and study design.

4. The relationship between COVID-19 and population movement is recursive but the authors have dealt on only one direction of this, and it should be clearly explained.

5. Good job with your analysis and pictorial illustrations.

Round 2

Reviewer 2 Report

Authors correctly addressed all the comments